

# *ν*-flows: Conditional neutrino regression

Matthew Leigh⋆, John Andrew Raine†, Knut Zoch and Tobias Golling

University of Geneva, Geneva, Switzerland

⋆ matthew.leigh@unige.ch , † john.raine@unige.ch

## Abstract

We present *ν*-Flows, a novel method for restricting the likelihood space of neutrino kinematics in high-energy collider experiments using conditional normalising flows and deep invertible neural networks. This method allows the recovery of the full neutrino momentum which is usually left as a free parameter and permits one to sample neutrino values under a learned conditional likelihood given event observations. We demonstrate the success of *ν*-Flows in a case study by applying it to simulated semileptonic $t\bar{t}$ events and show that it can lead to more accurate momentum reconstruction, particularly of the longitudinal coordinate. We also show that this has direct benefits in a downstream task of jet association, leading to an improvement of up to a factor of 1.41 compared to conventional methods.



# 1 Introduction

Collider physics experiments such as those at the Large Hadron Collider (LHC) [1] are at the forefront of studying the fundamental interactions of nature. General purpose detectors such as ATLAS [2] and CMS [3] are designed to measure nearly all stable particles produced in the high-energy proton-proton collisions. This means that they can be used to probe almost all aspects of the Standard Model of particle physics (SM). Reconstruction of these particles from base detector signals requires sophisticated algorithms and significant computing power. In recent years, deep learning algorithms have attracted significant attention and have been used for both kinematic reconstruction and identification for a wide variety of physics objects in these experiments. Some examples of successful applications include electron identification [4] and jet flavour tagging [5–7]. Advances in deep learning provide exciting new avenues for further improving the reconstruction performance of collider experiments.

Neutrino reconstruction requires a slightly different approach to that of jets and electrons. Neutrinos only couple to the weak nuclear force and typically do not interact with the detector material. They effectively escape from collider experiments without leaving any measurable signal. Instead, their presence is inferred from the momentum imbalance calculated from all visible particles in the plane perpendicular[1] to the beam pipe. This imbalance is known as the missing transverse momentum $\vec{p}_T^{\text{miss}}$, and it serves as an experimental proxy for the net transverse momentum of all undetected particles. There is no such experimental proxy in the longitudinal direction for proton-proton collisions as the initial momentum of the colliding partons is unknown. In events that produce more than one neutrino, accurate $\vec{p}_T^{\text{miss}}$ reconstruction still leaves the individual neutrino kinematics under-constrained.

Many analyses in collider physics investigate processes that involve neutrino production, and these could benefit from knowing the individual kinematics of final-state neutrinos. A prime example is the study of the top quark. The top quark decays almost instantaneously, and 99.9% of decays produce a $b$-quark and a $W$ boson. In approximately one-third of these cases, the $W$ boson decays leptonically, producing a final-state with a neutrino. The top quark is the heaviest particle in the SM which implies that it has the largest coupling to the Higgs boson. The value of its mass $m_t$ has a unique role in the stability of the electroweak vacuum due to its presence in the quadratic term of the Higgs potential [8]. Due to its almost instantaneous decay, it provides us with a unique opportunity to measure the properties of a bare quark. For many top quark measurements it is important to reconstruct the full $t\bar{t}$ system, including top quarks which decay leptonically via a $W$-boson. However, due to the unknown momentum of neutrinos in the final state this can be a source of mis-modelling of observables or poor reconstruction efficiency.

We introduce $\nu$-Flows, a machine learning approach to fully reconstruct the neutrinos produced in collisions from the missing transverse momentum and observed event kinematics. The approach taken in this work is that while many possible momenta values might be possible, they may not all be equally likely. Our method utilises conditional normalising flows [9,10] which exploits the latest developments in deep Bayesian learning to leverage observed information from the final-state and combine it with an inductive bias to restrict the likelihood over the possible neutrino momentum values. By sampling from this conditional likelihood, we obtain plausible estimates of the momenta for each undetected particle for each event, allowing us to reconstruct topologies that involve neutrinos.

We demonstrate the applicability of $\nu$-Flows in a semileptonic $t\bar{t}$ decay which has one neutrino in the final-state. We use estimates of the neutrino kinematics produced by $\nu$-Flows

---

[1]The coordinate system used in this work to describe collider experiment observables follows the convention of the ATLAS collaboration. The $x$-axis and $y$-axis lie perpendicular to the beam pipe while the $z$-axis is parallel. Pseudorapidity is defined as $\eta = -\ln(\tan\frac{\theta}{2})$, where $\theta$ is the polar angle from the beamline.

to reconstruct properties of the top quark and compare these to standard methods of neutrino momentum estimation. Furthermore, we assess the impact of using $\nu$-Flows in an analysis by quantifying the performance improvement in kinematic event reconstruction by solving the combinatoric jet-parton assignment to reconstruct the $t\bar{t}$ system. This analysis step is key in many analyses measuring differential production cross sections of $t\bar{t}$ events [11–14] and precision measurements of the top quark, for example the top quark mass in events containing a single lepton [15–18].

It is worth highlighting that, although focus is placed on neutrino reconstruction in $t\bar{t}$ events with a single lepton, the method can be adapted and applied to many other use cases. By changing the process used to train the model as well as the predicted neutrino multiplicity, $\nu$-Flows could be applied to many other processes, for example in the Higgs sector. In addition to neutrinos, many beyond the Standard Model (BSM) theories introduce new weakly interacting massive particles which are also expected to escape the detector without leaving any directly measurable signal. The $\nu$-Flows approach could also be used to determine their momenta. These applications are not studied in this work, however they demonstrate the variety of potential processes for which $\nu$-Flows could be of interest.

The source code[2] and data[3] used for this project are publicly available and can be found online.

## 2  Method

Estimation of neutrino momenta $\vec{p}^{\,\nu}$ from our set of visible particles can be framed as an inverse problem. The forward problem, which describes the transformation from $\vec{p}^{\,\nu}$ and other underlying variables to the observed quantities, is well understood and can be approximated by some stochastic process, such as the Monte Carlo simulations used in collider physics. But the inverse problem is difficult to approximate and the likelihood of the observations can only be implicitly defined by the simulation. The solution is also not unique; for example, due to the range of possible initial longitudinal momenta or the possibility of any number of multiple neutrinos. This is made even further complicated due to detector resolution effects. Standard deep learning regression methods collapse both the likelihood and posterior into a point estimate. This is undesirable as it gives no concept of solution diversity or uncertainty and ignores the fact that multiple solutions could exist. A probabilistic approach that can provide the likelihood over a range of viable solutions, rather than collapsing to just one, is required.

One promising method to perform full likelihood inference is to use conditional normalising flows. A normalising flow is a parametric diffeomorphism that defines a map between two probability densities over their respective spaces $f_\theta : X \to Z$. They typically map a complex probability distribution $p_X(x)$ into a simple density $p_Z(z)$ in a latent space with known properties, usually a multivariate normal distribution. These functions are often expressed using invertible neural networks (INNs) which are by design bijective, efficiently invertible, and possess a tractable Jacobian. Efficient density estimation under $X$ is obtained using the change of variables formula

$$p_X(x) = p_Z\big(f_\theta(x)\big)\Big|\det\big(J_f(x)\big)\Big|, \tag{1}$$

where $J_f(x)$ is the Jacobian of $f_\theta$ evaluated at $x$. This allows the generation of new data given $p_X(x)$ by sampling from $p_Z(z)$ and applying the inverse of the bijection $f_\theta^{-1}(z)$.

Normalising flows have seen great success in the field of computer vision for unconditional generation [20–22]. Conditional normalising flows use conditional invertible neural networks

[2]https://github.com/mattcleigh/neutrino_flows
[3]https://doi.org/10.5281/zenodo.6782987 [19]

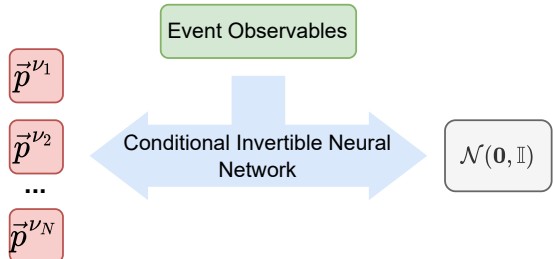

Figure 1: A schematic overview of the cINN used in $\nu$-Flows which predicts the momentum vector of $N$ many neutrinos as a condition of some chosen event observables. The latent density is chosen to be a multivariate normal distribution with $3N$ dimensions, $\mathcal{N}(\mathbf{0}, \mathbb{I})$.

(cINN) [23], defined by trainable parameters $\theta$, to incorporate contextual information $c$ into the map and lead to expressive conditional densities $p(x|c)$ when training with a maximum (log-)likelihood objective defined by

$$\arg\max_{\theta}\Big(\log\big(p_X(x|c)\big)\Big) = \arg\max_{\theta}\Big(\log\big(p_Z\big(f_\theta(x|c)\big)\big) + \log\Big|\det\big(J_f(x|c)\big)\Big|\Big). \qquad (2)$$

Our method for $\vec{p}^{\,\nu}$ likelihood estimation, called $\nu$-Flows, is built using cINNs. These types of networks have already been used in collider physics, with notable applications including event generation [24], anomaly detection [25–27], density estimation [28], detector unfolding [29], and detector simulation [30, 31].

$\nu$-Flows define a map from the combined space of all neutrino momenta to a simple density of equal dimension. To leverage information from the rest of the event, variables from event reconstruction are used as conditional inputs in the cINN. The flow can be trained directly to approximate the full conditional likelihood over the neutrino kinematics by performing gradient ascent on Equation 2. This leads to a rich description of the probability space, effectively allowing degrees of freedom to be recovered with interpretable uncertainties. A simplified diagram of this process is shown in Figure 1.

$\nu$-Flows can be applied to a wide variety of processes involving any number of invisible particles. However, for it to learn a useful likelihood it not only requires the observed information but also underlying assumptions or implicit biases. For example, the assumption of the number of neutrinos or non-interacting particles in the event is built into the structure of the cINN. Another necessary assumption is the underlying physical process being studied, which is ingrained into the flow by the composition and properties of the training set. Restrictions on the probability space of momenta are achievable by testing the probability of potential solutions under the observed kinematics of reconstructed physics objects in the event and the relationships between them given the assumed process. For each process or assumption, a specific implementation of $\nu$-Flows should be utilised because without leveraging these implicit biases it is not possible to constrain the possible phase space of solutions.

## 3 Case Study: Semileptonic $t\bar{t}$

In this work, we demonstrate an implementation of $\nu$-Flows applied to semileptonic $t\bar{t}$ decays. The final-state of this process contains at least four jets, a lepton, and a single neutrino. The goal is to use $\nu$-Flows to recover the $\vec{p}^{\,\nu}$, allowing us to fully reconstruct the whole $t\bar{t}$ system. Semileptonic $t\bar{t}$ events provide a logical starting point to introduce $\nu$-Flows and benchmark

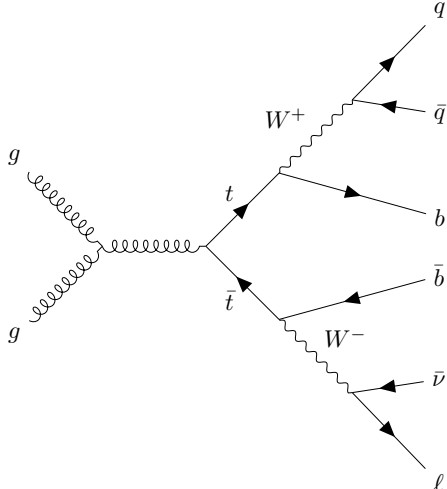

Figure 2: An example Feynman diagram showing one of the top quark pair production modes at the LHC, with one of the top quarks decaying into a final state containing a single lepton and neutrino.

their performance in comparison to standard techniques, before expanding to other topologies with more neutrinos and additional degrees of freedom.

A standard approach [13–18,32–34] to estimate $\vec{p}^{\,\nu}$ uses a kinematic constraint which can be expressed as

$$p_z^\nu = \frac{-b \pm \sqrt{b^2 - 4ac}}{2a}\,,\tag{3}$$

where

$$
\begin{aligned}
a &= (p_z^\ell)^2 - (E^\ell)^2\,,\\
b &= \alpha p_z^\ell\,,\\
c &= \frac{\alpha^2}{4} - (E^\ell)^2 (p_T^\nu)^2\,,\\
\alpha &= m_W^2 - m_\ell^2 + 2(p_x^\ell p_x^\nu + p_y^\ell p_y^\nu)\,.
\end{aligned}
$$

Here $p_x^\ell, p_y^\ell, p_z^\ell, E^\ell$ are the components of the four momenta of the lepton, and $m_\ell$ is its invariant mass (511 keV for electrons and 105.7 MeV for muons), $p_T^\nu$ is the transverse momentum of the neutrino, measured by $|\vec{p}_T^{\,\mathrm{miss}}|$, with x and y components $p_x^\nu$ and $p_y^\nu$. The mass of the $W$ boson is set to $m_W = 80.38$ GeV.

This approach has several drawbacks. Firstly, by assuming an exact value for $m_W$, any results or downstream tasks are biased, as it does not consider the natural width of $m_W$. Secondly, it assumes that the transverse momentum of the neutrino $p_T^\nu$ is perfectly captured by $\vec{p}_T^{\,\mathrm{miss}}$ and does not account for the misidentification, resolution, or mismodelling effects in the lepton or $\vec{p}_T^{\,\mathrm{miss}}$ reconstruction. These two effects can lead to Equation 3 yielding no real solutions. Here, the convention is to drop the imaginary component. An additional drawback is that even in the case where all objects are perfectly reconstructed, the equation can yield two real solutions. There is typically no strong reason to favour one solution over the other, though the result with the smaller magnitude is usually taken. Alternatively, both solutions are considered in any downstream tasks.

In contrast, $\nu$-Flows does not make such hard assumptions. From the composition of the training data, it can learn the width of the $m_W$ distribution and propagate that to a complex distribution over the longitudinal momenta. By providing $\nu$-Flows with additional information

from the event, it learns the probabilistic relationship between $\vec{p}_{\mathrm{T}}^{\,\mathrm{miss}}$, $\vec{p}^{\,\ell}$, and the target. With more contextual information, $\nu$-Flows combines observables in a fully probabilistic manner to learn the conditional distribution of possible solutions without collapsing the reconstruction down to singular values. Furthermore, while performance is expected to degrade, the architecture of $\nu$-Flows can be trivially scaled to predict any fixed number of neutrino momenta, it would just need to be retrained on the new process. In contrast, traditional approaches differ from one channel to another. For example the kinematic constraint method is not applicable in dilepton $t\bar{t}$ production where other techniques, such as Neutrino Weighting [35–37], are used.

## 3.1 Input Data and Targets

The data used in this work consists of simulated $t\bar{t}$ events where exactly one of the top quarks produces a b-jet and leptonically decaying $W^{\pm}$ boson. This corresponds to a final state containing either $(e, \nu_e)$ or $(\mu, \nu_\mu)$, or their corresponding antiparticles [19], as shown in Figure 2. All sets of events are generated from simulated proton-proton collisions at a center-of-mass energy of $\sqrt{s} = 13$ TeV.

Hard interactions are simulated using MadGraph5_aMC@NLO [38] (v3.1.0), with decays of top quarks and $W$ bosons modelled with MadSpin [39]. The mass of the top quark is set to $m_t = 173$ GeV for all events. The event generation is interfaced to Pythia [40] (v8.243) to model parton shower and hadronisation. All steps use the NNPDF2.3LO PDF set [41] with $\alpha_S(m_Z) = 0.130$, as provided by the LHAPDF [42] framework. The detector response is simulated using Delphes [43] (v3.4.2) with a parametrisation that mimics the response of the ATLAS detector [2]. Jets are reconstructed using energy-flow objects and the anti-$k_t$ algorithm [44] in the FastJet implementation [45] with a radius parameter of $R = 0.4$. Jet $b$-tagging corresponding to an inclusive signal efficiency of 70% is used to identify jets originating from $b$-quarks. Events are required to contain exactly one reconstructed electron or muon with $p_{\mathrm{T}} > 15$ GeV in the range $|\eta| < 2.5$ and at least four jets with $p_{\mathrm{T}} > 25$ GeV in the range $|\eta| < 2.5$. At least two of the jets are required to pass the $b$-tagging criteria. For truth labelling, jets were matched to partons within a radius of $\Delta R < 0.4$. Events containing jets matched to multiple partons were removed from the training and evaluation datasets. Around 600k events are used to train the model and an additional 100k events are used for evaluating performance.

Variables from event reconstruction are used as conditioning inputs to all models presented in this work. These include the kinematics of the signal lepton, kinematics and $b$-tagging information of the reconstructed jets, the $\vec{p}_{\mathrm{T}}^{\,\mathrm{miss}}$, and additional event observables. Up to 10 jets, as ordered by $p_{\mathrm{T}}$, are selected per event. The full set of inputs is described in Table 1. The target distribution for the networks is the single neutrino three-momentum vector defined by $\left(p_x^\nu, p_y^\nu, \eta^\nu\right)$. The coordinate system used to represent the momentum of each physics object, including the neutrino, was optimised as part of a hyperparameter scan, though there is not a strong dependence on coordinate choice. In this study using $\eta$ instead of $p_z$ was found to deliver the best performance, alongside the natural logarithm of the energy $\log E^j$ for the lepton and jets. The target density $p_Z(z)$ is chosen to be a standard normal distribution.

## 3.2 cINN Setup

The architecture of the $\nu$-Flows optimised for the neutrino in semileptonic $t\bar{t}$ decays is shown in Figure 3. The conditioning variables $c$ are first passed through a feed-forward (FF) network to ensure that the same high-level features are provided to each of the cINN blocks. In the FF component, a Deep Set [46] is used to extract information from the jets due to its ability to handle varying jet multiplicities while also remaining permutation invariant. The main cINN

Table 1: The different input observables used as conditional variables $c$ in the normalising flow.

| Category | Variables | Description |
|---|---|---|
| $\vec{p}_T^{\text{miss}}$ | $p_x^{\text{miss}}, p_y^{\text{miss}}$ | Missing transverse momentum 2-vector |
| Lepton | $p_x^{\ell}, p_y^{\ell}, \eta^{\ell}, \log E^{\ell}$ <br> $\ell^{flav}$ | Lepton momentum 4-vector <br> Whether lepton is an electron or muon |
| Jets | $p_x^{j}, p_y^{j}, \eta^{j}, \log E^{j}$ <br> $isB$ | Jet momentum 4-vector <br> Whether jet passes $b$-tagging criteria |
| Misc | $N_{\text{jets}}, N_{\text{bjets}}$ | Jet and $b$-jet multiplicities in the event |

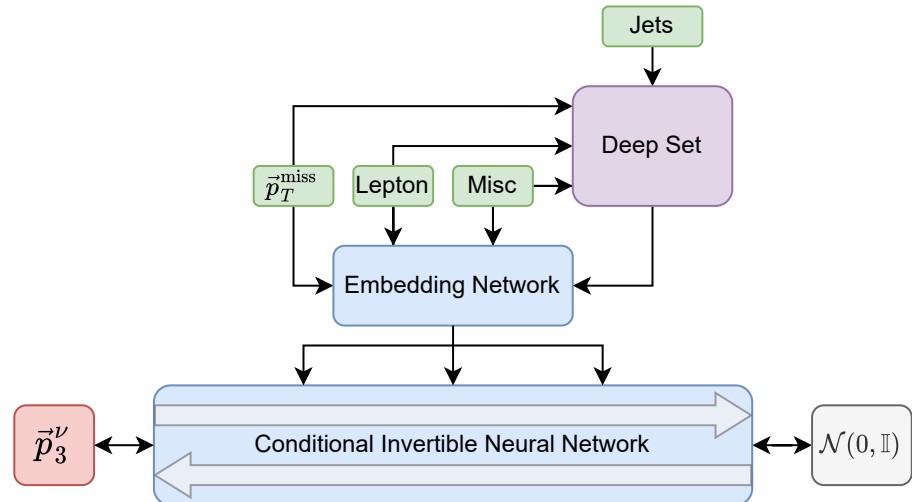

Figure 3: A schema of the $\nu$-Flows for semileptonic $t\bar{t}$. The four classes of conditioning inputs are shown in green and are used as inputs for both the Deep Set and the Embedding Network. There is only one neutrino in the event, so the input and output vectors of the cINN are three-dimensional.

blocks consist of seven rational-quadratic spline coupling layers [20]. Further details on the specific structure of each module can be found in Appendix A.

The cINN is trained on the objective function in Equation 2 using the Adam optimiser [47] with default $\beta$ parameters and a batch size of 256. We use a cosine annealing scheduler that cycles the learning rate from zero to $5 \times 10^{-4}$ and back every 2 epochs. Gradient clipping is essential for stable convergence and a max L2-norm of 5 is used. As a preprocessing step, all conditioning and target variables are independently normalised using the variance and mean of the training set. For cross-validation, 10% of the training dataset is reserved as a holdout set and early stopping is used with a patience parameter of 30 epochs. We use PyTorch [48] and nflows [49] to construct and train the cINN.

## 3.3 Feed-Forward Network

For comparisons of performance, we train a separate standard regression network that follows the same structure as the FF component of $\nu$-Flows but with a deeper embedding network used to predict the neutrino three-momentum directly. The FF network is trained using the

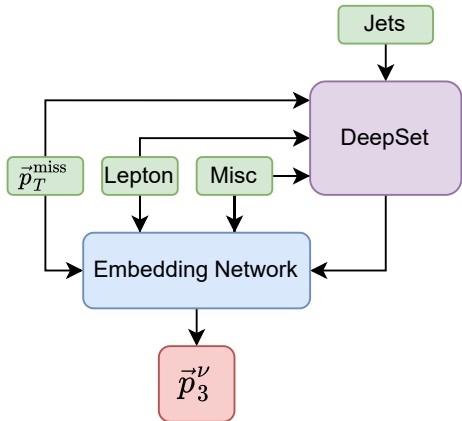

Figure 4: A schema of the $\nu$-FF network for semileptonic $t\bar{t}$. It uses the same input and target variables as $\nu$-Flows, but it trained using standard supervised regression methods.

Smooth-L1 loss function [50], with $\vec{p}^{\,\nu}$ as the target variable. We use the same training data, optimiser, learning rate scheduler, gradient clipping, and early stopping method as $\nu$-Flows. This method is referred to as $\nu$-FF and a schematic overview of its architecture is shown in Figure 4.

## 4  Performance

The $\nu$-Flow ($\nu$-FF) network was trained using an NVIDIA GeForce RTX 2080 Ti and the minimum validation loss was reached after approximately four (two) hours. Single event inference for one neutrino as measured on an AMD Ryzen 5900Hx is $\mathcal{O}(20\,\mathrm{ms})$. For a single event, multiple solutions can be calculated with the flow in parallel, and multiple events can be processed as a batch, resulting in faster inference times over a full dataset.

For $\nu$-Flows, two different configurations for conditional neutrino reconstruction are investigated. Both approaches use the same normalising flow trained on $t\bar{t}$ events. $\nu$-Flows(sample) represents the case where a single neutrino is sampled per event using the conditional probability density learned by the flow. This method of sampling is less biased but suffers from a high variance. As an alternative we also introduce $\nu$-Flows(mode) to stochastically approximate $\arg\max_x p_X(x|c)$. This is done by conditionally generating 256 neutrinos per event and keeping the one with the highest probability evaluated using the change of variables formula in Equation 2.

These methods are compared to the current standard approach which uses $\vec{p}_{\mathrm{T}}^{\,\mathrm{miss}}$ and Equation 3, as well as to the prediction from $\nu$-FF. As an upper benchmark, we compare all methods to using the true values of the neutrino momenta taken from the simulation. Plots labelled *Truth* refer only to using the true neutrino values, and all other properties, like those of the leptons or the jets, are taken from the reconstructed objects.

To best illustrate the benefits of a probabilistic method such as $\nu$-Flows, Figure 5 shows the reconstruction of the neutrino pseudorapidity for three different samples drawn from the evaluation dataset using the $m_W$ constraint method, $\nu$-FF, and $\nu$-Flows. In Figure 5(a) the true value of $\eta^{\nu}$ is around $-1.70$. One of the solutions of the $m_W$ constraint method is close to the true value and is around $-1.55$ while the other is significantly further away at $-3.05$. There is no indication *a priori* which of these two solutions will be closer to the truth and this is one of the main drawbacks of the method. $\nu$-Flows on the other hand provides us

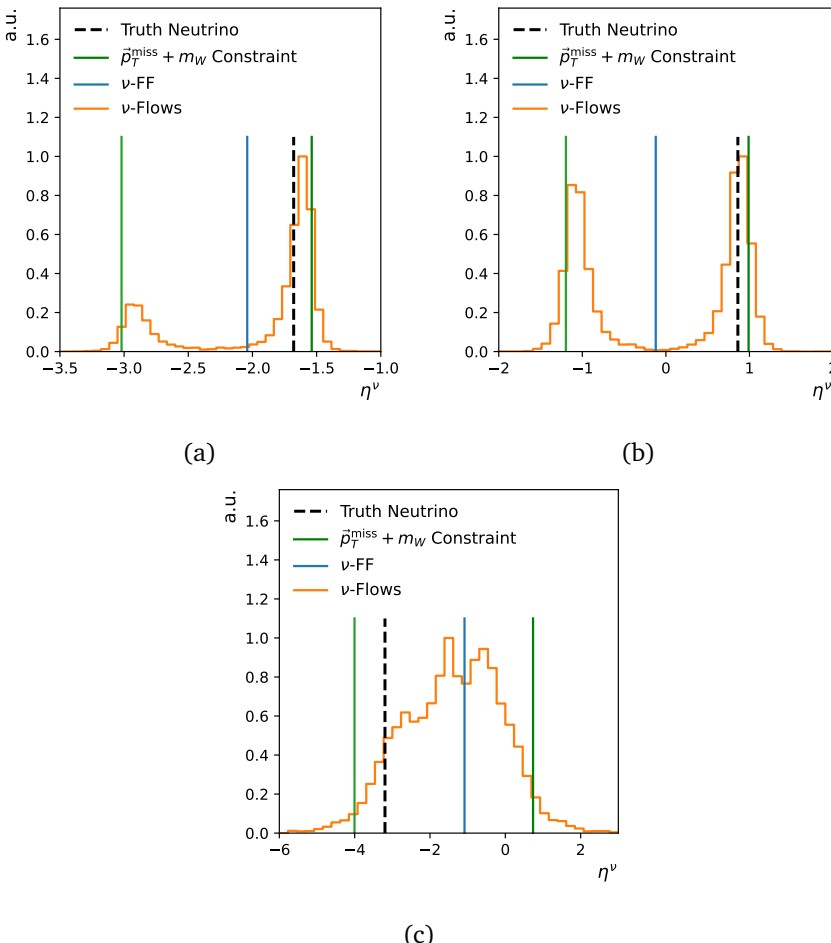

Figure 5: The pseudorapidity ($\eta$) of three different neutrinos selected from the evaluation dataset. The true values are shown in black. The two solutions from the $m_W$ constraint method are shown in green. The single point estimate using $\nu$-FF is shown in blue. The $\eta$ marginal for full conditional probability density learned by $\nu$-Flows is shown in orange. The $\nu$-Flows(sample) method corresponds to taking a single random sample under the conditional probability distribution and $\nu$-Flows(mode) corresponds to taking the most probable solution, which is equivalent to choosing the value at the peak of the distribution.

with the full probability across a range of $\eta^\nu$ values and shows a distribution with two local peaks corresponding to the quadratic solutions. This is worth noting as $\nu$-Flows was able to relearn the kinematic relationship detailed in Equation 3 entirely from data. But unlike the $m_W$ constraint solutions, $\nu$-Flows gives us interpretable uncertainties.

We also trained a version of $\nu$-Flows using quadratic solutions as extra conditioning inputs and observed a slight performance increase. However, we felt that the version which had to relearn this relationship purely from the dataset better demonstrated the power and expressiveness of the method. Furthermore, using $\nu$-Flows without the quadratic solutions also meant the same architecture can be applied to final-states with multiple neutrinos, where the quadratic method would be invalid.

For the event represented by 5(a), $\nu$-Flows indicates a preference for one of the possible solutions, with the highest localised cumulative distribution occurring at $\eta^\nu \approx -1.60$, close to the true value. In contrast, $\nu$-FF results in a point estimate close to $-2.05$ which falls between

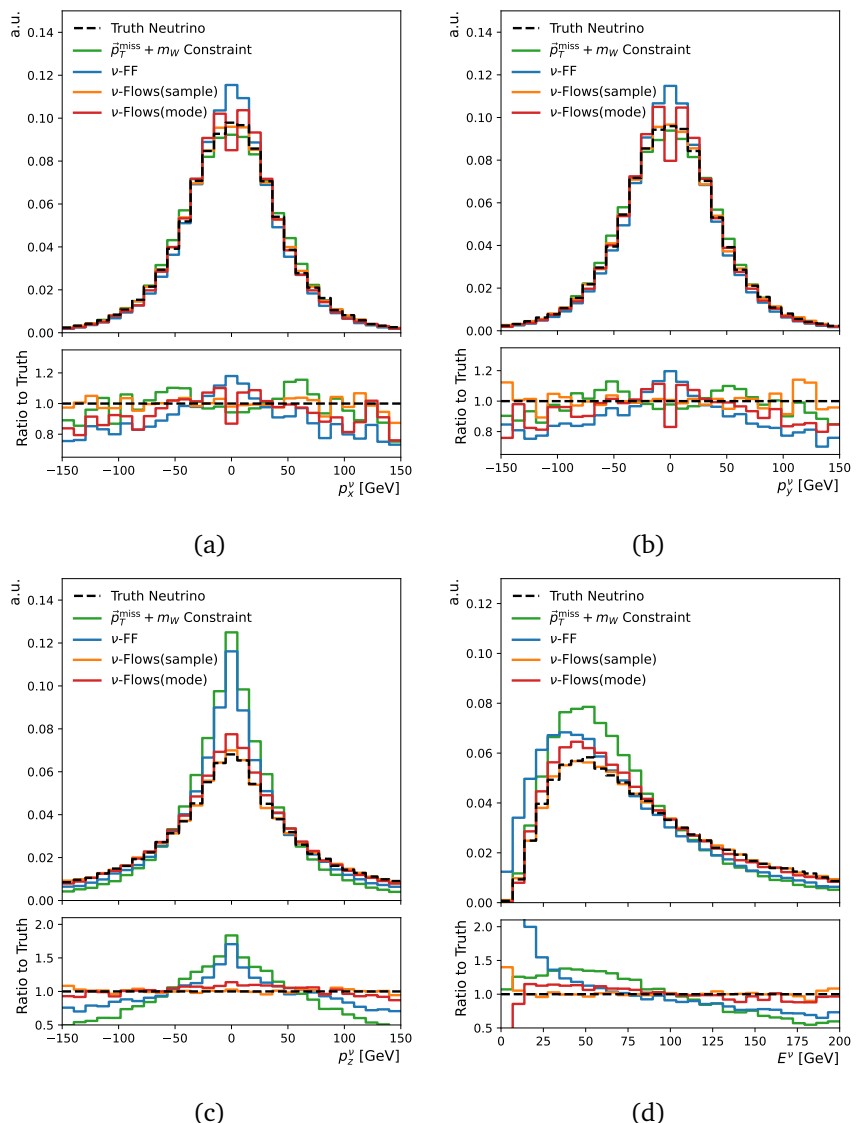

Figure 6: Distributions of each component of the neutrino four-momentum using the different reconstruction methods.

the two peaks, an area of low probability as estimated by $\nu$-Flows. It was observed that the $\nu$-FF predictions were almost identical to taking the average of the 256 samples generated by the flow. This is expected as the symmetrical loss function used to train $\nu$-FF collapses the posterior towards its centroid value.

Figure 5(b) shows a similar situation where $\nu$-Flows reproduces the multimodal probability distribution as expected by the kinematic constraint but with less of a preference for one solution over the other. Because of this $\nu$-FF results in a point estimate close to the average of the two solutions, resulting in an estimate much closer to $\eta^\nu \approx 0$.

Figure 5(c) shows an event where none of the methods could provide a good estimate for $\eta^\nu$. For all methods, including the mass constraint, to fail similarly points to an overall poor reconstruction of the objects in the event, namely $\vec{p}_T^{\text{miss}}$ and the single lepton. We still wish to further investigate specific failure cases, but it is important to note that the relative width or uncertainty displayed by the likelihood plot of $\nu$-Flows has increased correspondingly. This shows another benefit of this probabilistic approach as it can identify this event as being poorly reconstructed and one can filter it from downstream tasks.

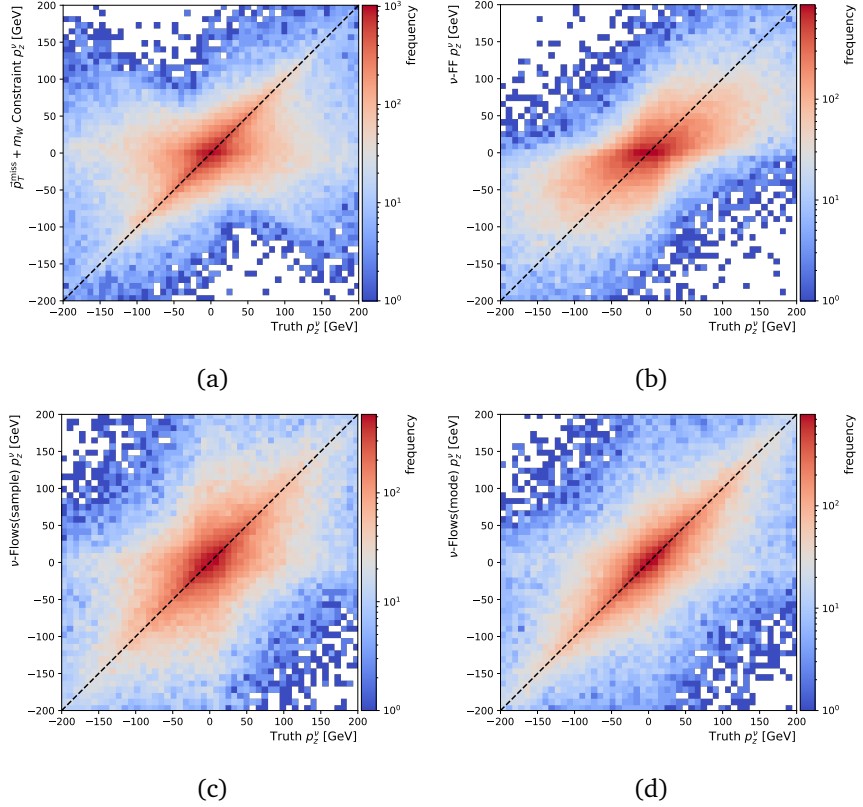

Figure 7: Two-dimensional histograms showing the reconstructed versus true $p_z^\nu$ using both solutions of the $m_W$ kinematic constraint (a), $\nu$-FF, (b), $\nu$-Flows(sample) (c), and $\nu$-Flows(mode) (d). The diagonal line represents ideal reconstruction.

The distribution of the neutrino four-momentum using the different methods for reconstruction are shown in Figure 6. For all coordinates, the distribution of the $\nu$-Flows(sample) is closest to the true momentum distribution. The $\nu$-FF and $m_W$ constraint methods induce a negative bias towards zero. This is most notable for $p_z^\nu$, shown in Figure 6(c), where both methods significantly overestimate the fraction of events close to zero. The negative bias in $\nu$-FF is caused by the model often guessing between the two kinematic solutions, as shown by Figure 5. This results in an underestimation of the energy as shown by Figure 6(d). $\nu$-Flows(mode) also possesses a negative bias in $p_z^\nu$ and $E^\nu$, although it is not as significant. There are notable artefacts in the $\nu$-Flows(mode) distributions in the transverse plane which causes a double peak around 20 GeV. This is caused by the shape of the $p_x$ and $p_y$ distributions of the jets and leptons, which due to the cut on $p_T$ also exhibit these double peaks.

Figure 7 shows heatmaps of 2D histograms using coordinates defined by the reconstructed and true $p_z^\nu$. Once again the bias towards zero is apparent in the $m_W$ constraint solutions and in the $\nu$-FF, both with an overestimation at zero. Both $\nu$-Flows models show a good correlation to *Truth*, however $\nu$-Flows(sample) suffers from a higher variance, showing the drawback in taking a single sample from the learned density. Here $\nu$-Flows(mode) shows good performance with the bulk of events being highly correlated with the true values while also showing no obvious bias.

The reconstructed invariant mass of the leptonic $W$ is shown in Figure 8(a), calculated using the momentum vector of the reconstructed lepton and each estimate of $p_z^\nu$. The distribution using the true neutrino is almost exactly matched by $\nu$-Flows(sample), while $\nu$-Flows(mode) is tightly centered around the mean. $\nu$-FF shows a notable offset of the mean by around 6 GeV.

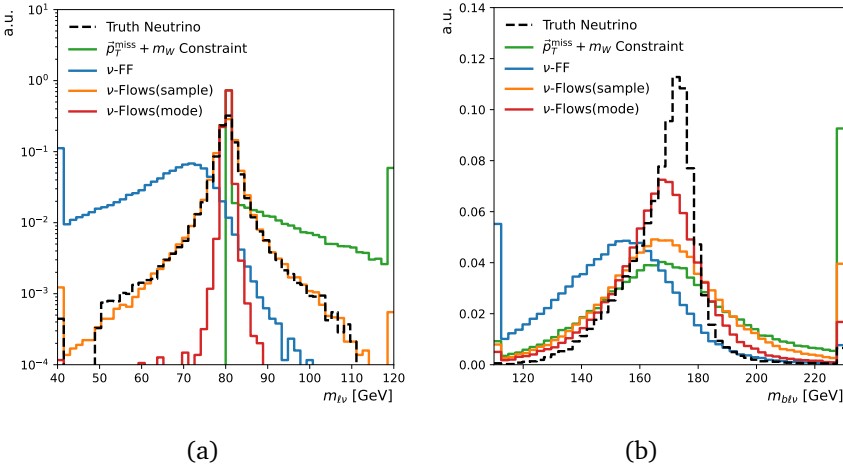

(a)            (b)

Figure 8: Distributions of the invariant mass of the $\ell\nu$ (a) and $b\ell\nu$ (b) systems using different neutrino reconstruction methods. All methods use reconstructed variables for the lepton and jet kinematics and *Truth Neutrino* uses the true neutrino.

The $m_W$ constraint results in nearly all events having exactly $m_{\ell\nu} = 80.38$ GeV, as expected, and the positive tail arises from events which lead to no real solutions for Equation 3. As is expected, $\nu$-Flows(mode) is biased towards the central value of the $m_W$ since it is estimating the most likely neutrino, which is therefore coupled with the most likely value for $m_W$.

When looking at the correlation between the reconstructed $m_W$ values and the true values, no correlations are observed for any of the methods. We find that the resolution effects in the $\vec{p}_T^{\,\text{miss}}$ are enough to destroy all information about the $m_W$ of the event. This is shown in Figure 15. This observation holds even when using the true value $p_z^\nu$ alongside $\vec{p}_T^{\,\text{miss}}$. It is worth noting that $\nu$-Flows learns the distribution of $m_W$ across the dataset even though it could not specify it on an event-by-event basis. This further demonstrates that it has learned to restrict its predictions of $p_z^\nu$ to the true space of possible solutions.

The reconstructed invariant mass of the leptonic top quark is shown in Figure 8(b). The correct $b$-jet from the leptonically decaying top quark is used in the calculation of the top mass. This is done to highlight the effect of the neutrino reconstruction, and thus only events for which the $b$-jet is reconstructed are shown. The $\nu$-FF method produces a shifted mass distribution, demonstrating a strong negative bias, with its peak at around 155 GeV. All other methods reduce this bias, but still peak at around 169 GeV, slightly under the simulated top mass of 173 GeV. Notably, the top mass distribution produced when using the true neutrino is negatively skewed while all other distributions are more symmetrical. The $m_W$ constraint method produces the distribution with the largest variance, resulting in a significant number of events with a reconstructed top mass greater than 230 GeV as shown by the overflow bin. The $\nu$-Flows(sample) method reduces this mass variance to around the same level as $\nu$-FF but without the negative shift. The $\nu$-Flows(mode) method further reduces this variance and produces the mass distribution most similar to *Truth*.

## $\chi^2$ Jet Association

To assess the impact of $\nu$-Flows in an analysis, we investigate its impact on a common downstream task, solving the combinatoric assignment of jets to final-state partons in semileptonic $t\bar{t}$ events. Solving the combinatoric assignment is a key component of a wide range of top quark physics analyses, from measurements of the top quark mass [15–18], (differential) cross section measurements of $t\bar{t}$ production [11–14], to measurements of spin correlation [51] and

charge asymmetry [34] in $t\bar{t}$ events.

Initially, it is unknown which (if any) of the jets that were observed in the event can be associated with the $b$-quark which was produced alongside the leptonically decaying $W$ boson ($b_{lep}$). In the final-state of the semileptonic $t\bar{t}$ channel there are four partons originating from the $t\bar{t}$ decay. These are the $b$-quarks from the leptonically and hadronically decaying top quarks ($b_{lep}$ and $b_{had}$ respectively), as well as the two decay products from the hadronically decaying $W$ boson, $q_1$ and $q_2$. Additional jets are also reconstructed from initial state radiation, final-state radiation, and pileup interactions. One of the most common methods used to assign the reconstructed jets to each parton is the $\chi^2$ fit [52]. The jet-assignment derived using this method is dependent on the neutrino kinematics, thus it can be used to demonstrate the benefits of having a more accurate neutrino estimate.

It is important to note this is just one of many jet combinatoric solving methods. Another popular approach is KLFitter [53] which is similarly dependent on the neutrino momentum. More recent approaches use machine learning to perform the associations [54–59] and have shown significant performance gains over the $\chi^2$ method. All of these combinatoric techniques should be complemented by $\nu$-Flows, though we demonstrate the potential gains using the $\chi^2$ method as it is already widely used in analyses [52, 60–62].

In the $\chi^2$ fit method, every possible jet permutation is tested, and the one with the lowest $\chi^2$ value defined by

$$\chi^2 = \frac{(m_W - m_{\ell\nu})^2}{\sigma_{\ell\nu}} + \frac{(m_W - m_{qq})^2}{\sigma_{qq}} + \frac{(m_t - m_{b\ell\nu})^2}{\sigma_{b\ell\nu}} + \frac{(m_t - m_{bqq})^2}{\sigma_{bqq}} \tag{4}$$

is kept. In this work, the $\sigma$ values are taken from the root-mean-square error of the relevant mass distributions, using the true jet-assignments, and are derived for each neutrino reconstruction method separately. We perform the $\chi^2$ fit using permutations of up to 9 leading $p_T$ ordered jets and record the parton association accuracy for each neutrino reconstruction method.

The $b_{lep}$ matching efficiency has the highest dependence on the neutrino in the $\chi^2$ fit and the association accuracy of the $b_{lep}$ is shown in Table 2. Using estimates from either $\nu$-Flows(sample) or $\nu$-Flows(mode) results in an improved matching efficiency compared to the standard kinematic approach. The $\chi^2$ fit performed with estimates from $\nu$-Flows(mode) instead of the $m_W$ constraint led to an increase in accuracy by a factor of 1.03 for events with four jets and 1.41 for events with nine jets. For events with a low number of jets, few permutations exist, which means that the neutrino term is less likely to have an impact in Equation 4.

Table 2: The fraction of events for which the $\chi^2$ method identified the correct $b_{lep}$ jet using the various neutrino estimation methods. The results are binned by the number of reconstructed jets in the event. Events must first pass a selection requirement where the partons were reconstructed as jets, so a correct permutation was at least possible. This selection did not change the ranking of the methods.

| Neutrino Type | Number of Jets | | | | | |
| --- | --- | --- | --- | --- | --- | --- |
| | 4 | 5 | 6 | 7 | 8 | 9 |
| Truth Neutrino | 0.864 | 0.753 | 0.686 | 0.641 | 0.611 | 0.587 |
| $\vec{p}_T^{\,\text{miss}}$ and $m_W$ Constraint | 0.790 | 0.576 | 0.476 | 0.398 | 0.366 | 0.286 |
| $\nu$-FF | 0.754 | 0.533 | 0.410 | 0.353 | 0.300 | 0.302 |
| $\nu$-Flows(sample) | 0.803 | 0.624 | 0.515 | 0.457 | 0.391 | 0.357 |
| $\nu$-Flows(mode) | **0.813** | **0.664** | **0.575** | **0.508** | **0.481** | **0.405** |

Therefore, the observed relationship between the performance gained using $\nu$-Flows and the number of jets in the event is expected. By improving the jet to parton matching efficiency the measurements of $t\bar{t}$ event properties will be of direct benefit, and as a result $\nu$-Flows can be expected to bring improvements to a range of measurements, however future studies will be needed to confirm these expectations.

## 5 Conclusions

We introduce $\nu$-Flows, a probabilistic model for conditional neutrino momentum estimation. We show that in semileptonic $t\bar{t}$ events $\nu$-Flows leads to better overall momentum reconstruction in comparison to both standard kinematic approaches and deep feed-forward networks. This in turn leads to an improvement in the downstream task of jet-parton assignment, as demonstrated using the $\chi^2$ method for solving the jet associations in $t\bar{t}$ events, a key component in many top quark analyses. More sophisticated algorithms for jet-assignment that use deep learning [58] have been shown to be very successful and may combine well with $\nu$-Flows.

It is interesting to note the relationship between the regression accuracy and the jet-parton assignment. When training the flow with full access to the truth parton labels for each jet, performance was observed to increase. When removing the jets as inputs to the network entirely, the performance is observed to decrease. This indicates a cyclic dependency, whereby the jet-parton assignment and the neutrino estimation both improve each other. A combined training approach with multiple tasks could be an avenue of further study.

The performance of $\nu$-Flows remains to be demonstrated in additional final-states, including those with more than one neutrino and therefore under-constrained transverse momenta. However, the architecture should be trivial to extend to these final states. A natural extension to the processes studied in this work is dileptonic $t\bar{t}$ decays. Furthermore, the full density produced by $\nu$-Flows contains more information than just a single neutrino solution, and could itself be used to reject events where the conditional probability is insufficiently constrained.

## Acknowledgements

ML, JR, KZ and TG are supported through funding from the SNSF Sinergia grant called Robust Deep Density Models for High-Energy Particle Physics and Solar Flare Analysis (RODEM) with funding number CRSII5_193716. ML is further supported with funding acquired through the Swiss Government Excellence Scholarships for Foreign Scholars, and KZ is funded through a Feodor Lynen Research Fellowship from the Alexander von Humboldt foundation.

## A Network Structure

### Conditional Attention Deep Set

Several methods for extracting variables from the jet container were studied in the development of $\nu$-Flows. These included manually extracting specific global variables from the jet container, as well as flattening the $p_T$ ordered set and passing this tensor through a dense network. We found that the Deep Set, specifically with attention pooling, performed considerably better.

Our Deep Set contains three dense networks, the `Feature Net`, the `Attention Net`, and the `Final Net` as shown in Figure 9. The jet variables from Table 1 are passed separately

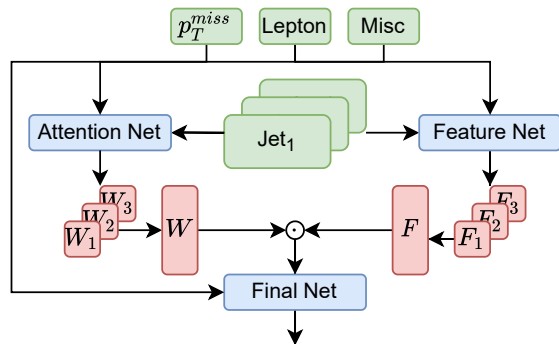

Figure 9: The attention weighted Deep Set for the jet container.

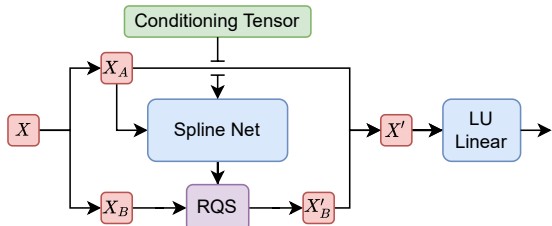

Figure 10: The building blocks of the conditional invertible neural network in $\nu$-Flows.

through the `Feature Net` to extract representations per jet $f_i$, and separately through the `Attention Net` to extract a weight per jet $w_i$. We then combine these outputs to perform a weighted sum of the representations of the $N$ jets in each event.

$$F = \sum_{i}^{N} w_i \cdot f_i \,.$$

The result is then passed through the `Final Net` to obtain the extracted features of the entire jet container. Conditional information from the $\vec{p}_{\mathrm{T}}^{\mathrm{miss}}$, lepton, and Misc variables are provided to each of the dense networks by concatenating them together with the jet inputs. The `Attention Net` produces a positive definite weight by applying an exponential activation function in the final layer.

**cINN Layer**

Many different configurations for the cINN were tested over the course of this work. Combining conditional coupling layers, with rational-quadratic spline transformers [20], and Lower-Upper triangular (LU) decomposed linear layers resulted in the best-observed performance at reconstructing the neutrino three momenta. This block is shown in Figure 10. The cINN is constructed of seven alternating coupling layers. In the very first coupling layer of the flow, we split the neutrino three-momentum by selecting the transverse coordinates for $X_A$ and the longitudinal coordinate for $X_B$. We then alternate this splitting with each subsequent coupling layer. We found that the masking order did have an impact on the final performance. Conditioning information is provided to the network by concatenating the extracted high-level features from the FF module to the inputs of the `Spline Net`. The python package *nflows* is used to construct the cINN.

**Dense Network Hyperparameters**

The $\nu$-Flows model in Figure 3 contains 5 different types of dense network. The three networks in the Deep Set, an `Embedding Network`, and a `Spline Net` in each layer of the cINN. The hyperparameters were determined by several grid searches using reconstruction performance on a validation set. All dense networks have two hidden layers of 64 nodes each. Each hidden layer applies the LeakyReLU [63] activation function with a slope parameter of 0.1 and Layer-Normalisation [64]. Additive residual connections are used between each hidden layer. Conditional information is injected into the dense networks by concatenating the context tensors to the inputs.

The $\nu$-FF network uses the same structure as the FF component of $\nu$-Flows but with an `Embedding Network` with 4 hidden layers and an output layer with three nodes, corresponding to the neutrino three-momentum.

## B  Additional Plots and Tables

Table 3: The various $\sigma$ values measured in GeV for use in the $\chi^2$ fit shown in Equation 4. These were calculated using the true jet associations and root-mean-square error from the top and $W$ boson masses, set to $173\,\text{GeV}$ and $80.38\,\text{GeV}$ respectively.

|  | $\sigma_{\ell\nu}$ | $\sigma_{b\ell\nu}$ | $\sigma_{qq}$ | $\sigma_{bqq}$ |
|---|---|---|---|---|
| Truth Neutrino | 4.67 | 17.33 | | |
| $\vec{p}_\text{T}^{\text{miss}}$ and $m_W$ Constraint | 31.11 | 50.92 | | |
| $\nu$-FF | 15.32 | 25.99 | 18.07 | 27.17 |
| $\nu$-Flows(sample) | 5.64 | 33.67 | | |
| $\nu$-Flows(mode) | 1.28 | 24.80 | | |

Table 4: The fraction of events for which the $\chi^2$ method identified the correct $b_{had}$ jet using the various neutrino estimation methods. The results are binned by the number of reconstructed jets in the event. Events must first pass a selection requirement where the partons were reconstructed as jets, so a correct permutation was at least possible.

| Neutrino Type | Number of Jets | | | | | |
|---|---|---|---|---|---|---|
|  | 4 | 5 | 6 | 7 | 8 | 9 |
| Truth Neutrino | 0.647 | 0.540 | 0.457 | 0.384 | 0.392 | 0.278 |
| $\vec{p}_\text{T}^{\text{miss}}$ and $m_W$ Constraint | 0.618 | 0.521 | 0.439 | 0.381 | 0.357 | 0.270 |
| $\nu$-FF | 0.591 | 0.492 | 0.417 | 0.358 | 0.355 | 0.302 |
| $\nu$-Flows(sample) | 0.619 | 0.518 | 0.436 | 0.364 | **0.358** | 0.286 |
| $\nu$-Flows(mode) | **0.621** | **0.522** | **0.444** | **0.382** | 0.353 | **0.278** |

Table 5: The fraction of events for which the $\chi^2$ method identified the leading $q_{1,2}$ jet using the various neutrino estimation methods. The $\chi^2$ method is invariant under a permutation of $q_1$ and $q_2$. The results are binned by the number of reconstructed jets in the event. Events must first pass a selection requirement where the partons were reconstructed as jets, so a correct permutation was at least possible.

| | Number of Jets | | | | | |
|---|---|---|---|---|---|---|
| Neutrino Type | 4 | 5 | 6 | 7 | 8 | 9 |
| Truth Neutrino | 0.707 | 0.626 | 0.558 | 0.490 | 0.470 | 0.325 |
| $\vec{p}_\mathrm{T}^\mathrm{miss}$ and $m_W$ Constraint | 0.690 | 0.613 | 0.547 | **0.490** | 0.442 | 0.349 |
| $\nu$-FF | 0.674 | 0.589 | 0.527 | 0.456 | 0.451 | **0.373** |
| $\nu$-Flows(sample) | 0.692 | 0.613 | 0.544 | 0.472 | **0.458** | 0.349 |
| $\nu$-Flows(mode) | **0.697** | **0.614** | **0.548** | 0.474 | 0.440 | 0.349 |

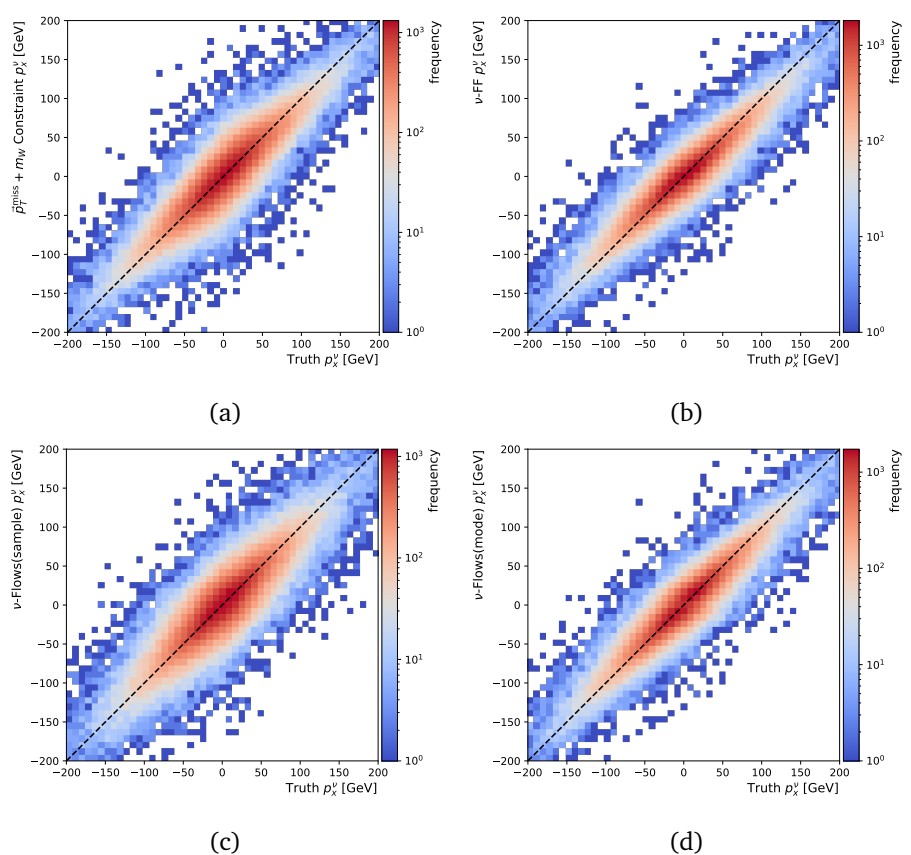

(a)

(b)

(c)

(d)

Figure 11: Two-dimensional histograms showing the reconstruction performance of $p_x^\nu$ using both solutions of the $m_w$ kinematic constraint (a), $\nu$-FF, (b), $\nu$-Flows(sample) (c), and $\nu$-Flows(mode) (d). In each plot, the true value is plotted along the x-axis and the reconstructed value is plotted along the y-axis. The diagonal line represents ideal reconstruction. The $p_y^\nu$ distribution results were virtually identical to these.

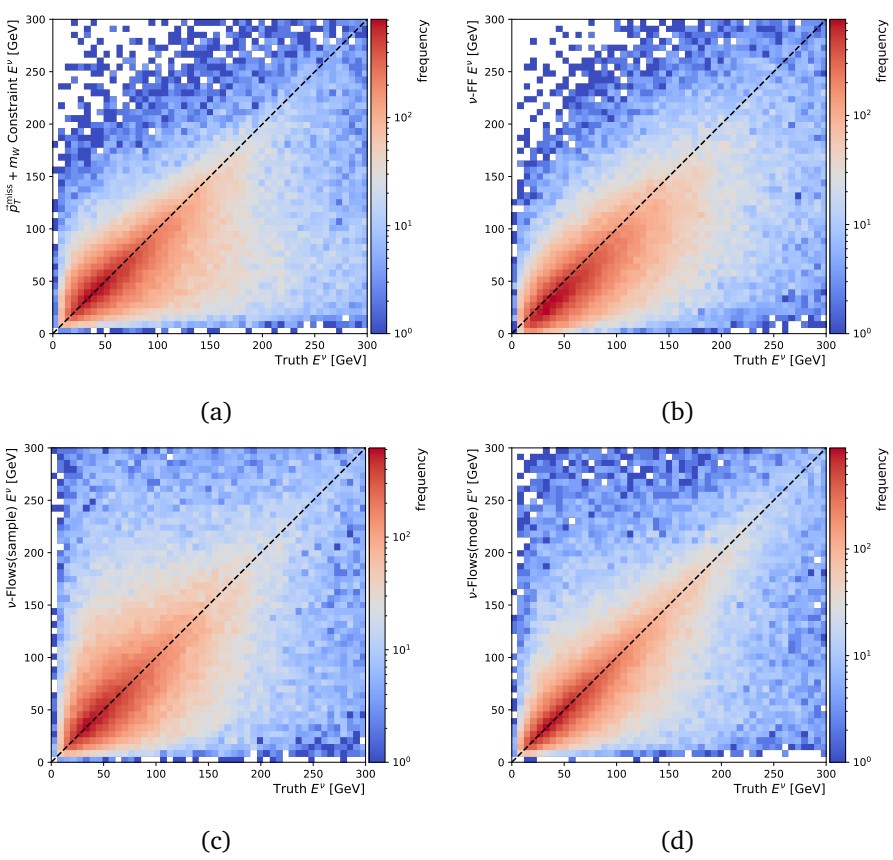

Figure 12: Two-dimensional histograms showing the reconstruction performance of the neutrino energy using both solutions of the $m_W$ kinematic constraint (a), $\nu$-FF, (b), $\nu$-Flows(sample) (c), and $\nu$-Flows(mode) (d). In each plot, the true value is plotted along the x-axis and the reconstructed value is plotted along the y-axis. The diagonal line represents ideal reconstruction.

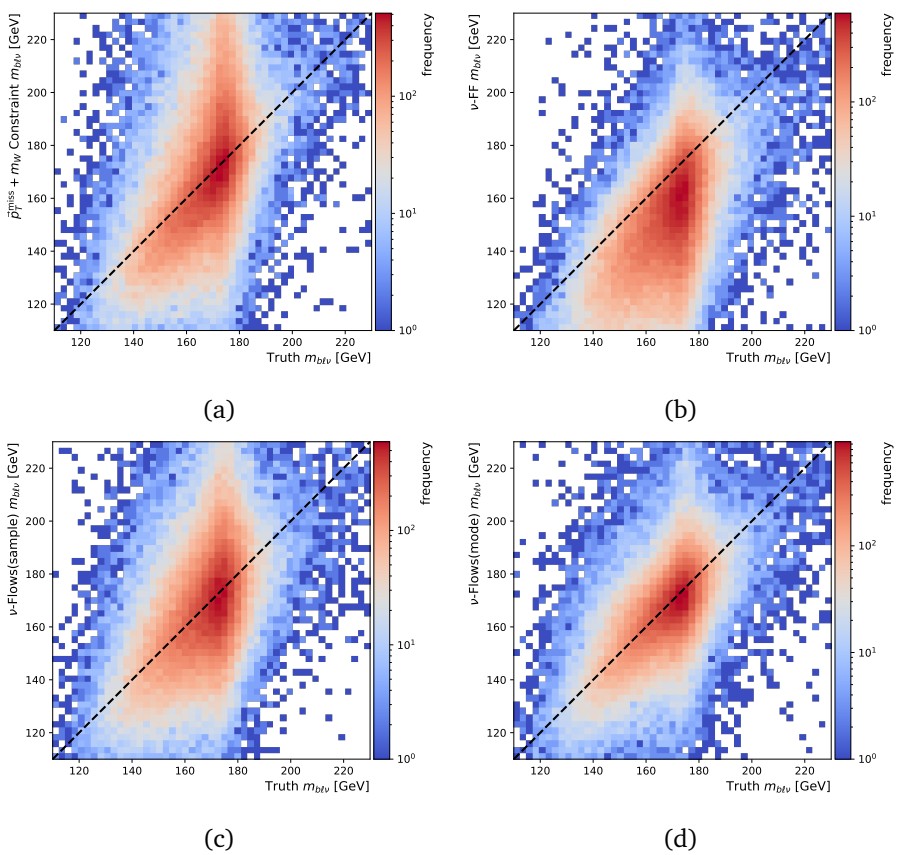

Figure 13: Two-dimensional histograms showing the reconstruction performance of the $t_{lep}$ mass using both solutions of the $m_w$ kinematic constraint (a), $\nu$-FF, (b), $\nu$-Flows(sample) (c), and $\nu$-Flows(mode) (d). In each plot, the true value is plotted along the x-axis and the reconstructed along the y-axis. The correct $b$-jet is used. The diagonal line represents ideal reconstruction.

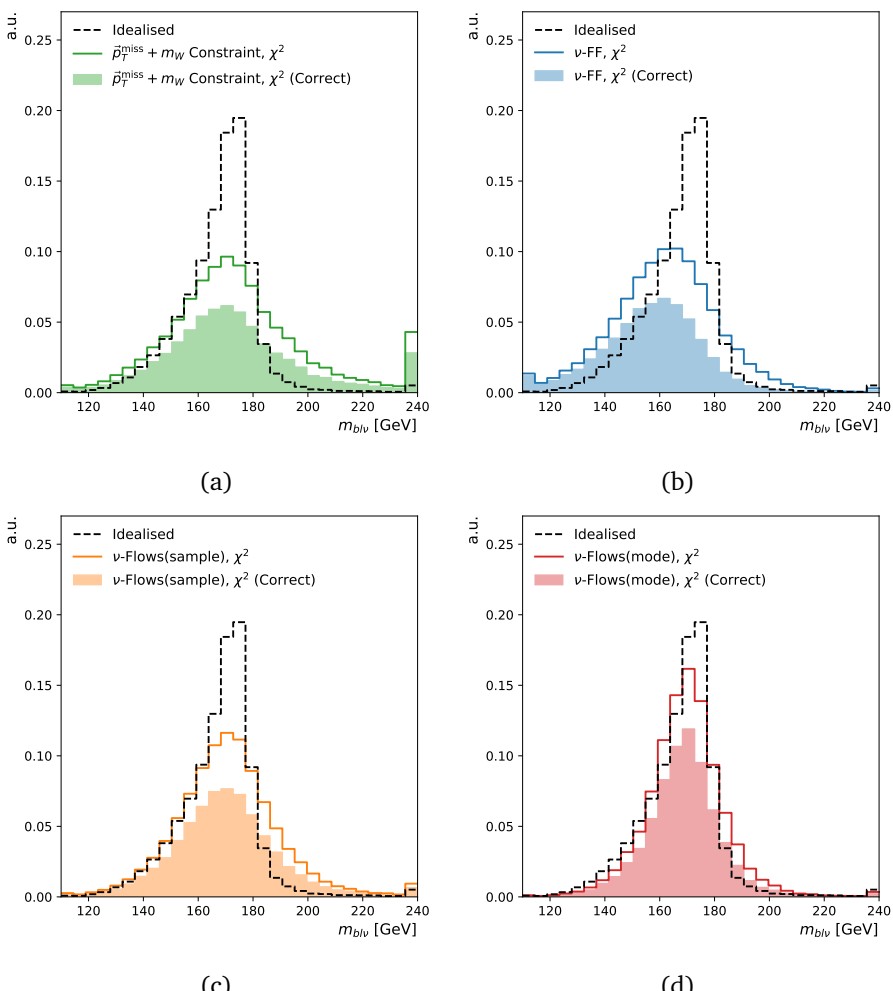

Figure 14: The reconstructed invariant mass of $b_{lep}$ using (a), $\nu$-FF, (b), $\nu$-Flows(sample) (c), and $\nu$-Flows(mode) (d). In each colored plot the $b$-jet is selected using the $\chi^2$ method. The *Idealised* curve uses both the true neutrino and the correct $b$-jet. The shaded plots show the subset of data for which the $\chi^2$ method identified the correct $b$-jet.

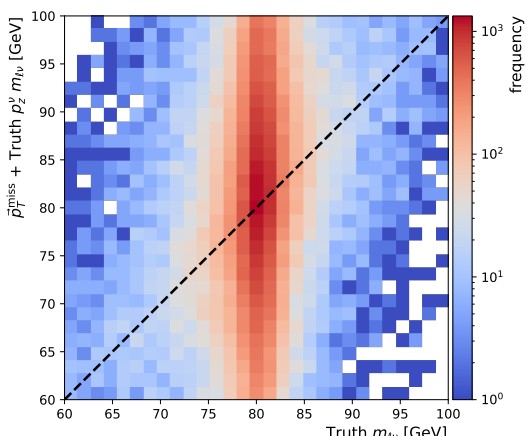

Figure 15: Two-dimensional histogram showing the reconstruction performance of the *W* boson mass using the missing transverse momentum combined with the Truth $p_z^\nu$. This illustrates how the resolution of the $p_T^{\text{miss}}$ reconstruction removes almost all correlation to the truth mass, and as such is a poor measure of how well the kinematics of a neutrino has been reconstructed.

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
