# Peer review of "$\nu$-Flows: Conditional Neutrino Regression"

_SciPost Physics, doi:SciPost Phys. 14, 159 (2023)_

## Round 1 · Referee Report · Anonymous · 2022-9-28

Strengths
1- introduces a completely novel method for regressing missing components in collider experiments
2- demonstrates improvement in a downstream task utilising the predictions of the method
3- all figures are overall clear and well presented
Weaknesses
1- uses only one very basic example case study of single lepton ttbar. Many claims about the ability of the method to scale up to more complex scenarios are made, but without such examples, it is not clear that the method will perform well in under-constrained systems .
2- the downstream task is an important one, but stops short of demonstrating an improvement on a final measurement, such as on the top quark mass
2- some minor grammatical issues can affect the readability of the paper, though it rarely affects the overall message
Report
Overall I found the paper to be a very interesting read and it shows very promising results in the simple example studied. The method is completely novel, utilising a relatively new ML architecture and applying it to a problem that has not seen similar work before.
It is held back by its restriction to this simple case and some of the claims about scaling should be revised in order not to oversell what has been demonstrated so far. It would also be desirable to see the effect of the improvements on a measureable quantity, such as the top quark mass. Nonetheless, the paper is certainly of high quality and I look forward to follow up works.
Detailed comments and questions can be found below. A further detailed editorial pass to review miscellaneous grammar issues, such as stub sentences, would also be a good idea.
Requested changes
Introduction
1- "...momentum must sum to zero." -> This is an overly strong statement that leaves out detector resolution effects or ambiguities about incoming parton directions, please revise
2- the Higgs boson examples are motivating but not followed up in the studies presented, and therefore seem out of place, particularly in coming before the top examples.
3- "...the possible phase space..." -> I find this phrase to bit somewhat unclear, consider revising
Method
4- I am confused by the definition of the term "ill-posed" here. Why do you say the 2-nu final state is an example, instead of the case you study here?
5- the parameter theta is not explicitly defined in the text
6- "our work falls under unfolding" -> is this really unfolding? consider rewording this sentence
7- "by default agrees with direct measurements" -> Isn't the point that it is impossible to do a direct measurement?
8- "designed and applied for" -> "designed for and applied t" or just "applied to"
Case Study
9- "produced frequently...relatively high efficiency" -> these are very vague statements and I am not sure what value they add here. Either be more precise, or remove
10- It would be great to add a "Feynman diagram" type figure to show what the decay chain and final state looks like. In the text it is assumed that the reader knows what the leptonic W decay is (ie W->lnu), but this is never defined
11- "at least two bjets, two other jets" -> this is ambiguous and probably should be inverted?
12- Please cite [26] earlier in the text, when introducing the method here. It may also be helpful to cite a ttbar example, such as https://arxiv.org/abs/1806.05463
13- please define all of the terms in these equations, and state mW=80.38 in the text
14- "several drawbacks" -> only one such drawback, ie the fixed mass, is discussed
15- Is nu-Flows really learning the resolution of the lepton kinematics? Consider revising this statement
16- It is stated that nu-Flows can scale to multiple neutrinos. While this seems true in principle, it has not been demonstated that the method can provide sensible results in these cases. Consider revising this statement
17- "decaying leptonically to a bjet..." -> this is a confusing statement, it is the W boson that decays leptonically - please revise
18- is there a b-jet cut applied in the event selection? I assume not since it is not stated, but this is unusual in ttbar analyses, please be explicit and motivate this choice
19- "These include...event observables" -> remove comma after "kinematics"
20- I presume one is selecting the 10 jets with highest pT, but this is not explicitly stated - please do so. And why do you limit to 10 jets here? Couldn't the DeepSets method take all jets in the events?
21- What is the motivation for regressing px and py, but eta instead of pz? Can a performance comparison be added here, if it differs? If the regression depends strongly on the representation of the desired quantity, this is an important detail that should receive discussion.
22- "For cross-validation, 10% of the training is..." -> "For cross-validation, 10% of the training dataset is..."
23- Please add some discussion of the training time, inference speed, memory overhead etc of the network
Performance
24- What exactly is the likelihood used to select the best prediction in the "mode" variation? Can this be plotted for the 256 predictions in the example events?
25- nu-FF details seem out of place - perhaps move them to a dedicated subsection of Sec3
26- "true values of the neutrino" -> "true values of the neutrino momenta"
27- "may be due to several reasons" -> only one such reason is discussed, please elaborate
28- were studies performed to check the validity of the deepset identifying the blep hypothesis, such as performing the reconstruction first and inputting only the ttbar system to the network? This statement suggests a cyclic dependence that may motivate further studies of a combined neutrino prediction and jet-parton assignment method, if it can be confirmed
29- "it can be proposed" -> this seems like it could trivially be checked and discussed in more detail here
30- further studies on the poorly reconstructed event(s) would be good to see. Is there some common features of these events - for example, are they always high (or low) (b-) jet multiplicity?
31- Similarly, it would be good to see a study that explicitly does the suggested removal of poorly estimated events, and check the effect this would have on the jet-parton assignments
32- "the negative bias in nu-FF is..." -> this sentence should be moved to the previous paragraph, and a similar sentence for the baseline is missing
33- "too high a variance" -> "a higher variance"
34- Could the specific values used to plot Fig5 also be added to Fig3 for the "mode" variation?
35- "when looking at correlations of mW" -> It would be good to have these plots at least in backup material.
36- Some of the figures, especially 3/4, would be difficult to read in greyscale or if the reader is colorblind. It would be god to at least try to make this easier (of course I realise this is difficult to achieve without worsening the general readability - I leave this to the authors disgression).
37 - Figure 5 (+9,10,11) has no z-axis label
38- all of the citations for the chi2 method are in the all-hadronic channel. Is there an example where this has been used in leptonic decays?
39- Some additional references for jet-parton assignment with deep learning are missing eg https://arxiv.org/abs/2010.09206 and https://arxiv.org/abs/2012.03542
40- Please include the fitted values of each sigma somewhere for reproducibility
41- How is the truth assignment defined for the chi2 method? Please describe this
42- It would be nice to also see numbers for bhad and Whad in the results. Although these are second order effects, they are in principle non-zero, and improvements here would help further sell the method.
Conclusions
43- "significant improvements in downstream tasks" -> "in the downstream task of jet-parton assignment"
References
44- one minor typo in reference 7, "collaboration" -> "Collaboration"

---

## Round 1 · Referee Report · Anonymous · 2022-11-9

Strengths
1) The nu-Flow method that the authors introduce yields a posterior distribution of the neutrino momenta that is better in two ways than traditional techniques: provides full likelihood information (good for multiple solutions and error estimation) and is less biased.
2) The article is well-written and pedagogical. The example used in the text is good to use cases and illustrative. The figures are well crafted.
Weaknesses
1) The introduction and earlier parts of the article overpromise given what the article provides.
2) The authors talk about multiple neutrino reconstruction, but only deal with one neutrino.
3) The authors talk in the beginning about how this can improve e.g. top mass measurements. This is not only not demonstrated, but I believe that this method cannot improve the final physics inference. As the authors nu-Flow is learning the correlations that already exist in the Monte Carlo.
Report
This is a nice article to read. I thought the discussion and the ideas were interesting; however, the article overpromises and underdelivers. I believe this can be solved by editing the introduction and conclusion appropriately.
My biggest concern is that I am not sure about the usefulness of this tool. The code is learning the posterior distribution of the neutrino momenta from Monte Car for a specific process/selection. As the authors show this matches the true distributions. But why not just draw the true distributions from the Monte Carlo directly and convert this to posteriors? This seems that should provide the same answers as the author's code, but in a simpler fashion.
Requested changes
1) Authors need to make the introduction/conclusion more appropriate to what they are actually doing in this article. Reduce slightly the things that could be done, but have not been shown to work, and state what has been done.
2) The authors need to provide a convincing argument and/or scenario where we are indeed improving some physics parameter measurement. Just being able to plot the neutrino momenta posterior distribution does not seem sufficient.

---

## Round 2 · Referee Report · Anonymous · 2023-1-11

Report

I find the author much improved manuscripts and the details replied to the comments received.

First of all, I agree with the authors statement that obtaining the posterior distribution of the neutrino conditional on a high-dimensional space is indeed not trivial and is a problem that your method solves well. However, I still believe that further (quantitative) evidence needs to be presented to claim that this improves the physics reach of ongoing analysis.

With the current draft rescoping, as manifested by the improved Introduction and Conclusion sections, I believe that this article is now suitable for publication.

---

## Round 2 · Referee Report · Anonymous · 2023-1-13

Report

Overall I find v2 of this paper to be a significant improvement in terms of language and presentation, with most of my comments addressed.

I would still like to see more on the poorly reconstructed events; eg, some basic checks on if there are common kinematic features in these events, in order to understand better what the network is/isn't learning. Additionally, the discussion the authors included in their response regarding the improvements from including additional jets, giving the truth jet-parton assignments, etc are interesting studies that I think are worth discussing in the paper.

Nonetheless, I think the paper is already of publishable quality.

Requested changes

p3: "of potential process" -> "of potential processes"

p6: The deltaR requirement for truth matching is stated in the text as 0.2, but in the authors response as 0.4 - check the paper is the right number :) Out of interest, how many events were removed due to ambiguous matching, and how many were removed due to incomplete matching?

p6: "This is in contrast to traditional approaches
where different approaches need to be optimised for final states with other multiplicities of
neutrino s in the final state, for example Neutrino Weighting [35–37] in the case of dilepton
t ̄t production"
-> this sentence is messy. I think what is trying to be conveyed is that "there are specific methods for specific neutrino multiplicities"? Since nu-flows would be re-optimised for whatever use-case (as stated in the sentence at the end of section 2), I think this sentence should be revisited. Again, since no attempt is made at final states with >1 neutrino, I think one should be careful on overpromising extensions because it is unclear how performant it will be.

p9: Please add discussion of the mass constraint method shown in Fig 5. The observation that v-FF more or less gives the average of the v-Flows method is interesting; I almost wonder if a combination of the v-FF and constraint methods would be able to give a prediction of similar quality to the full v-Flows method by using the distance from the v-FF to the two quadratic solutions to weight them, and what this would imply about what v-Flows is really learning.

p11: I find the text on fig7 to be very difficult to read - can the figures be made a bit larger to improve this? Fig 11,12,13,15 are better in that way, though increasing the text size slightly in all of these would also be good

p12: Some additional references for jet-parton assignment with deep learning are still missing eg https://arxiv.org/abs/2010.09206 and https://arxiv.org/abs/2012.03542

references: check for author/collaboration formatting, eg [37] is "T. C. Collaboration" :)

---

## Round 2 · Author Response

We would like to thank the reviewers for their very useful feedback and suggested changes. We have incorporated and addressed all points in the latest draft of the manuscript, and believe that they have made this a far stronger manuscript, in particular in the introduction and conclusions.

Following on from requests and questions we have the following general comments in reply to requests and comments.

We have been asked to provide a convincing argument or scenario that we are improving physics parameter measurements and not just able to plot the neutrino momentum posterior distributions.
We would like to point out that as presented in the results section, we not only study the conditional posterior distributions but also the impact on the event reconstruction efficiencies with the chi2 approach.
Combinatoric jet-parton assignment is a key component of a wide range of top quark analyses at ATLAS and CMS for measuring e.g. (differential) cross-sections [1-4], top quark mass [5-8], charge asymmetry [9] and spin correlation [10].
By being able to quantify the improvement in the reconstruction efficiency in the chi2 method, we can show that the approach is valuable for a wide range of measurements rather than focussing on a single one, and without needing to shift the focus of the manuscript away from the core method.
As such we feel that this comment is already addressed in the manuscript, and these references are all included in the updated draft with emphasis on this part added to the manuscript.

[1] https://arxiv.org/abs/1610.04191
[2] https://arxiv.org/abs/1803.08856
[3] https://arxiv.org/abs/1908.07305
[4] https://arxiv.org/abs/2108.02803
[5] https://arxiv.org/abs/1507.01769
[6] https://arxiv.org/abs/1805.01428
[7] https://arxiv.org/abs/1810.01772
[8] https://arxiv.org/abs/1905.02302
[9] https://arxiv.org/abs/2208.12095
[10] https://arxiv.org/abs/1511.06170

We would also like to address the confusion that we are simply picking a neutrino from the posterior distribution of neutrino momenta. This is a conditional posterior distribution given the O(50) dimension input space and with permutation invariance over all the jets in the event. This is not something that trivially can be done by drawing from the monte carlo directly, as one would have to know the jet assignments first. Furthermore, one would need to preserve the full joint posterior distribution in that high dimension space, which could have areas of low statistical significance, in order to evaluate for all possible conditional values.
During development we found the inclusion of all final state objects critical to achieve the best performance, and so even learning this solely as a the joint distribution of neutrino and lepton kinematics would not be sufficient.
Furthermore normalising flows are very adept at learning conditional distributions and perform very well at interpolation.

Additionally, although we are confident that v-flows can be applied to final states with multiple neutrinos and that it will work, this has not been demonstrated in the work covered in this manuscript.
This is something we plan to demonstrate with future work expanding the applications of v-flows to more processes. For now, we have changed the statement in the paper to reflect that the architecture can be modified to any number of neutrinos, without making explicit statements of how well it will perform (except that being a harder challenge, it is expected to be slightly less performant than the one neutrino case).

For the v-flows architecture, although we restrict the architecture to the first 10 jets in the event in principle we do not need to and can absolutely go to higher multiplicities; the cut off at 10 was done at the data processing level.
However, this choice has an almost negligible impact as fewer than one permille of all events have 10 jets, and it is expected that even fewer would have had 11 or more.
Furthermore, we included the target coordinate system as part of the hyperparameter scan for the v-flows model in this work. The choice presented in the manuscript choice performed best of all, with px, py, pz the second best as measured with the RMSE to the truth and with the loss function of the flow.
All choices of coordinate systems still performed well, and it does not have a strong influence on the performance.

An additional point raised which we also think deserves further discussion is on the dependence of the deep sets in being able to "identify" the b-jet from the leptonic W, and whether performing reconstruction on the events first would help with reconstructing the neutrino, and how this might lead to a cyclical dependence.
This is a very good point and something that we indeed have been discussing amongst ourselves as we move to using this for more advanced methods of jet association.

We have run some tests originally that used the fixed jet ordering based on truth matching (and therefore telling the network which was the blep, bhad, etc) and this performed much better than our current setup, so this does suggest an improved identification of which jet comes from which parton would be very useful.
We have also tested the performance without including any jet information and with the DeepSets we find it does bring a gain in performance even without the network knowing which jet is which.
Our hope also is that with a DeepSets architecture the network is able to use the summation nature of the deep sets pooling to be akin to momentum conservation. As a result, we hope that this is a generalisable architecture which would not need further optimisation for other final states and underlying processes.
But indeed, there is a cyclic dependency, whereby the jet-parton assignment and the neutrino estimation both improve each other, and a combined training approach is indeed something we would like to investigate in future work. But for now we hope that the simpler architecture means that this approach would be the egg to come before the chicken.

A suggestion was made on investigating poorly reconstructed events. This is not something we have done yet but absolutely want to do as part of future studies. A further suggestion was made to perform a study that explicitly removes poorly reconstructed events.
We wholeheartedly agree with this suggested study, though for this initial paper we wanted to restrict the scope to the machine learning approach. We would choose to study potential downstream applications and interpretations in detail when applying this in a concrete setting with the inclusion of background processes and a target measurement.
Currently we are identifying different metrics for identifying “poorly estimated events”, for example using the variance of the 256 samples, though since the learned predicted kinematic distributions of good events are almost always multimodal we need to balance this against poor reconstruction with a single wide distribution.

Finally, the colour palette of the plots has been checked to be colour blind friendly for most forms of colour blindness, however with changes for greyscale we found the plots worse for general readability. As such we have left the colour scheme as it was.

Once again, we would like to reiterate our thanks for the feedback on this manuscript.
Best regards,
Johnny, on behalf of the authors

---

## Round 2 · List of Changes

Introduction

On request we have reduced the amount of potential studies in the introduction, however we prefer to keep the mention to potential applications to avoid confusion that v-flows is purely for top quark physics.
We also motivate the choice of chi2 as a quantitative measure of improvement thanks to v-flows with the addition of more references.

- "...momentum must sum to zero." -> Overly strong statement rephrased to be more in line with standard description:

“Instead, their presence is inferred from the momentum imbalance calculated from all
visible particles in the plane perpendicular to the beam pipe. This imbalance is known as
the missing transverse momentum…”

- Paragraph introducing potential applications has been reformulated to remove emphasis from processes not covered in the paper.

- "...the possible phase space..." -> improved the clarity by removing the “phase space” sentence and replaced it with:
“Any meaningful insight into the kinematics of non-interacting can be useful in a wide range of both SM measurements and BSM searches.”

Method

- We have changed the phrasing to remove the use of ill-posed, and focus on the sample in the case study rather than mention 2l case.

- Parameter theta now defined in the text as network parameters

- "our work falls under unfolding" -> removed explicit statement, as work is not necessarily unfolding

- "by default agrees with direct measurements" -> Removed the sentence, so that it is more coherent. The paragraph now reads:
“Restrictions on the probability space of momenta are achievable by testing the probability of potential solutions under the observed kinematics of reconstructed physics objects in the event and the relationships between them given the assumed process.”

Case Study

- "designed and applied for" -> "applied to"

- "produced frequently...relatively high efficiency" -> removed sentence

- Added a Feynman diagram of ttbar (semi-leptonic decay)

- "at least two bjets, two other jets" -> Removed ambiguity: "The final-state of this process contains at least four jets, two of which are required to be identified as b-jets, a lepton, and a single neutrino."

- Ref. [26] has been moved to the beginning of paragraph, including additional reference for ttbar example

- Defined all terms in the equation including mW in the text:
“Here pℓx , pℓy , pℓz , Eℓ are the components of the four momenta of the lepton, and mℓ is its invariant mass (511 keV for electrons and 105.7 MeV for muons), pνT is the transverse momentum of the neutrino, measured by pTmiss, with x and y components pνx and pνy. The mass of the boson is set to mW = 80.38 GeV.”

- Expanded on statement that quadratic equation approach has several drawbacks (previously only one given):
“This approach has several drawbacks. Firstly, by assuming an exact value for mW, any
results or downstream tasks are biassed, as it does not consider the natural width of mW. Secondly, it assumes that the transverse momentum of the neutrino pνT is perfectly captured by pTmiss and does not account for the misidentification, resolution, or mismodelling effects in the lepton or pTmiss reconstruction. These two effects can lead to Equation 3 yielding no real solutions. Here, the convention is to drop the imaginary component. An additional drawback is that even in the case where all objects are perfectly reconstructed, the equation can yield two real solutions. There is typically no strong reason to favour one solution over the other, though the result with the smaller magnitude is usually taken. Alternatively, both solutions are considered in any downstream tasks.”

And to later in the section

“…different approaches need to be optimised for final states with other multiplicities of neutrinos in the final state, for example Neutrino Weighting [28–30] in the case of dilepton ttbar production.”

- Revised statement that v-flows learns resolution of lepton kinematics to be more accurate:
“By providing ν-Flows with additional information from the event, it learns the probabilistic relationship between pTmiss , pℓ, and the target.”

- Revised statement on application of v-flows to final states with multiple neutrinos:
“... while performance is expected to degrade, the architecture of ν-Flows can be trivially scaled to predict any fixed number of neutrino momenta, depending on the chosen underlying process.”

- "decaying leptonically to a bjet..." -> Changed the statement to be:
“The data used in this work consists of simulated ttbar events where exactly one of the top quarks produces a b-jet and leptonically decaying W boson.”

- added an explicit statement on the b-tagging criteria of selected jets: “At least two of the jets are required to pass the b-tagging criteria.”

- "These include...event observables" -> remove comma after "kinematics"

- Add explicit mention that v-flows chooses top 10 jets ordered by pT (NB: this is not a strict requirement for the architecture but done for technical reasons)

- Clarify that coordinate system is part of hyperparameter scan, but that it has only a small impact on performance:
“The coordinate system used to represent the momentum of each physics object, including the neutrino, was optimised as part of a hyperparameter scan, though there is not a strong dependence on coordinate choice. In this study using eta instead of pz was found to deliver the best performance, alongside the natural logarithm of the energy logEj for the lepton and jets.”

- "For cross-validation, 10% of the training is..." -> "For cross-validation, 10% of the training dataset is..."

Performance
- Added information on training and inference time
“The nu-Flow (nu-FF) network was trained using an NVIDIA GeForce RTX 2080 Ti and the minimum validation loss was reached after approximately four (two) hours. Single event inference for one neutrino as measured on an AMD Ryzen 5900Hx is O(20ms). For a single event, multiple solutions can be calculated with the flow in parallel, and multiple events can be processed as a batch, resulting in faster inference times over a full dataset.”

- Clarify definition of v-flows(mode) and how it is determined. As the likelihood here is the same from the loss using the change of variables formula. We now refer directly to using Equation 2 to calculate this probability, and have removed use of jargon and unclear phrasing.

- Added clarification to Mode in the figure captions:
“The eta marginal for full conditional probability density learned by nu-Flows is shown in orange. The nu-Flows(sample) method corresponds to taking a single random sample under the conditional probability distribution and $\nu$-Flows(mode) corresponds to taking the most probable solution, which is equivalent to choosing the value at the peak of the distribution.”

- nu-FF details moved them to a dedicated subsection of Sec3

- "true values of the neutrino" -> "true values of the neutrino momenta"

- Removed statement on why there may be a preference for solutions. We have discussed this amongst ourselves and find that our initial suggestion was assuming too much from the architecture. In principle the deep sets learns how to extract the most useful information to the task from the jets and additional inputs.
In general, the preference comes from the training data themselves, as the learned probability distribution over the neutrino kinematics reflects the data seen during training.

- We have tested the proposal of v-FF being the mean of the 256 examples and have now stated this in the manuscript rather than hypothesising:
“We observe that the nu-FF predictions are almost identical to taking the average of the 256 samples generated by the flow. This is expected as the symmetrical loss function used to train nu-FF collapses the posterior towards its centroid value.”

- Added a statement on investigating poorly reconstructed events, as at yet these have not yet been understood (is it due to the architecture, lack of information, or fundamental to the events):
“An important avenue of future work is investigating the common features of the events with poor reconstruction.”

- "the negative bias in nu-FF is..." -> this sentence has been moved to the previous paragraph

- "too high a variance" -> "a higher variance"

- Added z axis labels to Figs 5, 9, 10, 11

- Addition of many more references for chi2 and neutrino determination

- Added clarification to text for truth assignment used in chi2 method:
“For truth labelling, jets were matched to partons within a radius of delta R < 0.4.
Events containing jets matched to multiple partons were removed from the training and evaluation datasets.”

Conclusions

The conclusion is also suitably shorter, removing reference to studies not performed. However, we directly address that we have not tested this approach on multiple neutrino systems or other final states.

"significant improvements in downstream tasks" -> "in the downstream task of jet-parton assignment"
References

- Fixed typo in collaboration -> Collaboration

- Additional references added

Apendix

- Added an extra plot in the appendix showing nearly no correlations between mW and met+truth pz.

- Added table for the sigma values used in chi2 method to the appendix.

- Added the matching efficiencies for the bhad and whad in a table

---

## Round 3 · Author Response

We would like to thank both reviewers for the valuable input and feedback. We have addressed all suggestions and implemented suggested corrections as well as drafted responses to particular issues that were raised.
This has helped polish the paper and bring it to an overall standard we are proud of.

Most changes are minor and listed in the following section, but some other points are addressed here.

A reviewer was interested to note how many of the events had to be rejected due to ambiguous jet-parton matching. Out of the number of events that pass the selection criteria (kinematics, multiplicities, tagging), approximately 9% were discarded due to ambiguous jet-to-parton matching. Out of those remaining, approximately 52% are fully matchable, i.e. all four quarks from the ttbar event are unambiguously matched to reconstructed jets.

We were requested to add extra discussion to the results shown in Fig 5 as well as address the idea that one could use the v-FF network to provide a weighting to the two kinematic solutions based on proximity.
This is a very interesting observation and suggestion, and isn’t something we had had in mind. It would certainly be an interesting avenue of study to improve the naive v-FF and mass constraint methods, however for this paper we prefer to remain focused on the flows for this paper without a deep dive into exactly what it learns. Certainly from the jets and labelling we know it is more than just the mass constraint that it learns. Furthermore, there would still be the instances where the mass constraint method leads to no real solutions, whereas in all instances v-Flows is able to provide a full prediction over the potential neutrino momentum solutions.

Best regards,
The authors

---

## Round 3 · List of Changes

Introduction

Fixed a typo that was pointed out and has been changed
old: of potential process"
new: of potential processes

Case Study

The text has been changed to to reflect the correct truth parton matching of R=0.4.
old: a radius of R < 0.2
new: a radius of R < 0.4

It was pointed out that the wording of the following text bit messy and it has been modified to be more direct
old: This is in contrast to traditional approaches where different approaches need to be optimised for final states with other multiplicities of neutrino s in the final state, for example Neutrino Weighting [35–37] in the case of dilepton ttbar production
new: Furthermore, while performance is expected to degrade, the architecture of v-Flows can be trivially scaled to predict any fixed number of neutrino momenta, it would just need to be retrained on the new process. In contrast, traditional approaches differ from one channel to another. For example the kinematic constraint method is not applicable in dilepton tt production where other techniques, such as Neutrino Weighting [35–37], are used.

Results

We have changed the text in the previous paragraphs to make it more clear where comparisons to the mass constraint method in Fig 5 are being made. As well as added the following paragraph to the discussion of Fig 5.
new: For all methods, including the mass constraint, to fail similarly points to an overall poor reconstruction of the objects in the event, namely ptmiss and the single lepton. We still wish to further investigate specific failure cases, but it is important to note that the relative width or uncertainty displayed by the likelihood plot of v-Flows has increased correspondingly.

We have increased the size of images with hard to read text.

Included extra references for machine learning approaches to jet-parton assignment. The requested references for Spatter and SAJA have been added along with:
Erdmann et al, J. Instrum. 14, P11015 (2019)
ATLAS, Phys. Rev. D 97, 072016 (2018)
ATLAS, Phys. Rev. Lett. 125, 061802 (2020)

Conclusions

We were asked to add to the discussion some of the comments we addressed in the previous round of changes, alluding to how including jet-parton assignment information into the network improves its performance. The following paragraph has been added to the conclusion.
new: It is interesting to note the relationship between the regression accuracy and the jet-parton assignment. When training the flow with full access to the truth parton labels for each jet, performance was observed to increase. When removing the jets as inputs to the network entirely, the performance is observed to decrease. This indicates a cyclic dependency, whereby the jet-parton assignment and the neutrino estimation both improve each other. A combined training approach with multiple tasks could be an avenue of further study.

References

We have performed a review of each reference and fixed those plus other small inconsistencies.

---

## Editorial Decision

published